# The Generation and Characterization of Monoclonal Antibodies against the MPXV A29L Protein

**DOI:** 10.3390/v16081184

**Published:** 2024-07-24

**Authors:** Wenlong Zhu, Mengjia Zhang, Mengdi Zhang, Ran Jing, Jiaru Zhou, Hua Cao, Changcheng Liu, Hongmei Zhu, Ahmed H. Ghonaim, Sherin R. Rouby, Wentao Li

**Affiliations:** 1National Key Laboratory of Agricultural Microbiology, College of Veterinary Medicine, Huazhong Agricultural University, Wuhan 430070, China; zhuwl@webmail.hzau.edu.cn (W.Z.); zhangmengjia0210@mail.hzau.edu.cn (M.Z.); mengdi@webmail.hzau.edu.cn (M.Z.); jingran@webmail.hzau.edu.cn (R.J.); jiaru@webmail.hzau.edu.cn (J.Z.); caohua@webmail.hzau.edu.cn (H.C.); changchengliu0813@163.com (C.L.); zhuhongmei@mail.hzau.edu.cn (H.Z.); 2The Cooperative Innovation Center for Sustainable Pig Production, Wuhan 430070, China; drcahmed91@gmail.com; 3Desert Research Center, Cairo 11435, Egypt; 4Department of Veterinary Medicine, Faculty of Veterinary Medicine, Beni-Suef University, Beni-Suef 62511, Egypt; shereen.rouby@vet.bsu.edu.eg

**Keywords:** monkeypox virus, A29L, monoclonal antibody, VACV

## Abstract

Mpox (formerly known as monkeypox) is a zoonotic disease caused by monkeypox virus (MPXV), a DNA virus belonging to the Orthopoxvirus genus, in the Poxviridae family. The disease constitutes a moderate risk to public health at the global level. The MPXV A29L protein plays a crucial role in coordinating virion assembly and facilitating important virus-host interactions. This study focused on the expression, purification, and recombinant protein synthesis of the A29L protein of MPXV using prokaryotic systems. Using hybridoma technology, we successfully generated the monoclonal antibodies (mAbs) 1E12 and 4B2, which specifically recognize the A29L protein. These mAbs were found to be suitable for use in indirect immunofluorescence assays (IFA), Western blotting, and immunoprecipitation (IP). Our investigation also revealed that mAbs 1E12 and 4B2 could detect the A27L protein, a homologous protein found in the vaccinia virus Western Reserve (VACV WR) strain, using IFA, Western blotting, and immunoprecipitation (IP). Using mAbs 1E12 and 4B2 as primary immunological probes, A27L protein expression was detected as early as 6 h postinfection with VACV WR, with increasing protein levels being observed throughout the infection. This study enhances our understanding of the protein structure and function of MPXV and contributes to the development of specific MPXV detection methods.

## 1. Introduction

Monkeypox is an emerging zoonotic disease caused by MPXV, a member of the Orthopoxvirus genus in the Poxviridae family, which also includes variola virus (VARV), vaccinia virus (VACV), and cowpox virus (CPXV) [1]. MPXV was first isolated in 1958 from a specimen of the macaca cynomolgus monkey species, which exhibited signs of illness and was brought from Africa to Denmark. However, the natural host of MPXV remains unknown, although it is hypothesized that wild rodents and other small mammals may be responsible [2,3]. In the 1970s, cases of MPXV in human populations were first detected in several African countries. Since then, the virus has become endemic on the continent, leading to frequent reports of monkeypox infections in humans. However, since May 2022, there has been a sudden and dramatic increase in the number of cases of MPXV, with the sequences of prevalent strains being similar to those of West African strains [4]. Consequently, the World Health Organization (WHO) declared the monkeypox outbreak a global health emergency and assessed the global public health risk of monkeypox as moderate [5,6,7,8].

The genome of MPXV is linear and approximately 197 kb in length, containing a highly conserved central region consisting of 101 kb. The genome is divided into 16 segments (A~P) that encode more than 200 proteins [9]. The virus manifests in two distinct infectious forms: the intracellular mature virus (IMV) and the extracellular enveloped virus (EEV). Each of these forms exhibits unique surface proteins and employs distinct mechanisms to infect host cells [1]. The A29L protein, encoded by the A29L gene in region A, is a vesicle membrane protein found on the surface of IMV particles and serves as a monoclonal antibody target for MPXV. As a homologue of the A27L protein in VACV, the A29L protein plays a role in viral attachment to the host–cell membrane and functions as both a transcription factor for gene transcription and an enzyme for viral replication [10]. Certain peptides from the A27L protein have been identified as T-cell epitopes for CD4^+^ and CD8^+^ cells. The epitope (21–49 aa) on the A29L protein is located adjacent to the binding region of glycosaminoglycans (GAGs) and has been demonstrated to be a known neutralization target capable of neutralizing viral particles and interfering with viral adhesion to the host–cell surface [11,12,13,14].

In this study, the MPXV A29L protein was expressed in *E. coli*, and a hybridoma cell line secreting mAbs against the MPXV A29L protein was generated through mouse immunization. The specificity of the mAbs was tested using IFA, IP, and Western blotting. Additionally, the expression pattern of the A27L protein during virus infection in vitro was determined. The results of this study provide new insights into the development and specificity of mAbs targeting the MPXV A29L protein.

## 2. Methods

### 2.1. Viruses, Cells, and Animals

The VACV WR strain (GenBank accession no. NC_006998.1) was maintained in our laboratory. Human embryonic kidney (HEK-293T) cells, baby hamster kidney 21 (BHK-21) cells, and HeLa cells were cultured in Dulbecco’s modified Eagle’s medium (DMEM) (Gibco, Waltham, MA, USA) supplemented with 10% fetal bovine serum (FBS) and 1% penicillin/streptomycin (100 μg/mL) (Invitrogen, Waltham, MA, USA) at 37 °C with 5% CO_2_. Murine myeloma (SP2/0) cells were cultured in RPMI 1640 medium supplemented with 20% fetal bovine serum (FBS) and 1% penicillin/streptomycin (100 μg/mL) (Invitrogen) at 37 °C with 5% CO_2_. Six- to eight-week-old BALB/c mice were obtained from the Laboratory Animal Center of Huazhong Agricultural University.

### 2.2. The Expression and Purification of the MPXV A29L Protein

Gene fragments encoding the full-length MPXV A29L protein (GenBank ID: OR350452.1) were subjected to codon optimization for expression in *E. coli* and cloned in-frame into the pET-28a prokaryotic expression vector. *E. coli* BL21 (DE3) was transformed with a pET-28a plasmid expressing the MPXV A29L protein. The bacterial cells were cultured in 1 L of Luria broth (LB) supplemented with 50 μg/mL kanamycin at 37 °C until the OD_600_ reached 0.6. Isopropyl β-D-1-thiogalactopyranoside (IPTG) was then added at a final concentration of 0.8 mM to induce protein expression at 37 °C for 4 h. The bacterial culture was pelleted at 6000× *g* for 10 min at 4 °C and lysed in 100 mL of PBS by sonication. The lysate was separated by centrifugation at 13,000× *g* for 30 min. Recombinant proteins were purified from the supernatant by metal affinity chromatography using Ni Sepharose 6 Fast Flow (GE Healthcare Life Sciences, Chicago, IL, USA). The fusion protein was analyzed by Western blotting or SDS-PAGE followed by Coomassie Brilliant Blue staining.

### 2.3. The Generation and Screening of mAbs against the MPXV A29L Protein

Animal experimental protocols were approved by the Institutional Animal Ethical Committee, Experimental Animal Center of Huazhong Agriculture University, China (HZAUMO-2024-0182). Equal volumes of purified recombinant A29L protein and Freund’s adjuvant (Sigma, St. Louis, MO, USA) were mixed and then injected into 6- to 8-week-old BALB/c mice. The mice were immunized at two-week intervals, and serum antibodies were detected by ELISA on day 7 after the fourth immunization. Mice with high antibody titers were used for the final booster immunization and cell fusion. Mouse splenocytes were harvested on day 3 after the final booster immunization and mixed with preprepared myeloma cells at a 10:1 ratio for cell fusion in the presence of 50% PEG (Sigma, St. Louis, MO, USA). Positive cell clones were screened by ELISA and IFA, and two rounds of subcloning were performed using the limiting dilution method. To obtain monoclonal antibodies, the antibody-secreting hybridoma cells were injected intraperitoneally into mice sensitized with liquid paraffin, the ascites fluid was collected after 7 days, and the monoclonal antibody was bound to Protein A+G Agarose and eluted with 50 mM glycine, pH 2.7.

### 2.4. ELISA

Purified A29L protein was diluted in 50 mM sodium carbonate/bicarbonate buffer (pH 9.6) and was added to the wells at 4 °C overnight at a concentration of 10 μg/mL and a volume of 100 μL per well. The wells were washed three times with TBST (20 mM Tris–HCl, 150 mM NaCl, 0.1% Tween-20, pH 8.0) between each subsequent step. After blocking with 1% BSA in TBS-T for 1 h at room temperature, the hybridoma supernatant or mouse serum was added to each well and incubated for 2 h at room temperature. Next, the wells were incubated with HRP-conjugated goat anti-mouse IgG (H+L) (ABclonal, Wuhan, China) for 1 h. The optical density at 630 nm (OD630) was measured using an ELISA reader (Tecan, San Jose, CA, USA).

### 2.5. Indirect Immunofluorescence Assay

BHK-21 cells were grown on 24-well plates for 2 days until they reached 80% confluence. Half of the wells were infected with VACV WR at a multiplicity of infection (MOI) of 0.1, and the other half were mock-infected. At 24 h postinfection, the cells were fixed with 4% paraformaldehyde for 15 min at room temperature and then permeabilized with 0.1% Triton-X-100 for 15 min at room temperature. The fixed cells were blocked with 1% bovine serum albumin (BSA, Biosharp, Beijing, China) in PBS for 1 h, followed by a 1 h incubation period with 100 μL of hybridoma supernatant or 50 μg/mL of the purified mAbs 1E12 and 4B2. The cells were washed with PBST and then incubated for 1 h at 37 °C with an Alexa Fluor 488-labeled donkey anti-mouse IgG antibody (1:500 dilution). The nuclei were stained with DAPI (Solarbio, Beijing, China). Samples were examined using the laser confocal microscope (Nikon, Tokyo, Japan).

### 2.6. Western Blot Identification of mAbs

BHK-21 cells were seeded in 12-well plates and grown to 80–90% confluence before being inoculated with VACV WR. After 24 h of infection, the cells were harvested, lysed, mixed with loading buffer, and heated in a metal bath at 95 °C for 10 min before being subjected to SDS-PAGE for protein separation. The 0.45 μm PVDF membrane was activated by soaking in methanol for 2 min, and the gel was transferred to the PVDF membrane at 100 mA for 40 min. The membrane was blocked with 5% nonfat milk in TBST for 2 h at room temperature. A monoclonal antibody (mAb) against the A29L protein, which was prepared in this study, was used as the primary antibody at a dilution of 1:1000 and incubated overnight at 4 °C on a rotary platform. HRP-conjugated goat anti-mouse IgG was used as the secondary antibody at a dilution of 1:5000, and the membrane was incubated for 1 h at room temperature on a rotary platform. After each of the above steps, the membrane was washed three times with TBST for 5 min at room temperature. The ECL reagent was added, and the membrane was exposed to an imager for detection.

### 2.7. Identification of mAbs by Immunoprecipitation

BHK-21 cells were seeded on a confocal dish and grown to a semiconfluent state before being inoculated with VACV WR. The samples were lysed with cell lysis buffer for Western blotting and IP (Beyotime, Shanghai, China) with protease inhibitor cocktail tablets (Biosharp, Beijing, China) for 10 min on ice. Protein concentrations were then measured using the Bicinchoninic Acid (BCA) Protein Assay Kit (Biosharp, Beijing, China). Next, 100 μg of protein sample was incubated with 2 μg of mAbs overnight at 4 °C, and the mixture was bound to Protein A+G for 4 h at room temperature. After incubation, the beads were washed five times. The samples were then analyzed by Western blotting.

### 2.8. Investigating A27L Protein Expression Using mAbs

To investigate A27L protein expression during VACV WR infection, BHK-21 cells were inoculated at an MOI of 0.1, and cell samples were collected at 3, 6, 9, 12, 15, 18, 21, 24, and 36 h postinfection. A27L protein expression was detected by IFA and Western blotting using the mAbs 1E12 and 4B2.

### 2.9. Expression of A29L on HEK-293T Cells

Gene fragments encoding the full-length MPXV A29L protein (GenBank ID: OR350452.1) were amplified by PCR and cloned in-frame into the pCAGGS mammalian expression vector. The plasmid was transiently expressed in HEK-293T cells and then analyzed using Western blotting, IFA, and IP.

## 3. Results

### 3.1. The Expression and Purification of the Recombinant MPXV A29L Protein

The MPXV A29L protein-coding gene was subjected to codon optimization for expression in *E. coli*. As depicted in Figure 1A, the amplification of the target product using MPXV A29L gene primers yielded a 368 base pair product devoid of any extraneous bands. Subsequently, the purified product was ligated into the expression vector pET-28a (linearized plasmid of 5195 bp). As shown in Figure 1B, the constructed recombinant plasmid pET-A29L was successfully expressed in *E. coli* BL21 (DE3). The expected molecular weight of the protein was approximately 18.3 kDa. The products expressed in the supernatant were purified using Ni Sepharose™ 6 Fast Flow and identified by SDS-PAGE. The reactivity of the purified protein was verified by Western blotting using an anti-His mAb (Figure 1C).

### 3.2. The Production and Characterization of the mAb against the MPXV A29L Protein

Hybridoma cells that can infinitely multiply in vitro were obtained by cell fusion technology, hybridoma cells that could secrete the mAbs (1E12 or 4B2) against the MPXV A29L protein were selected by combining the limiting dilution method with ELISA and IFA, and then the mAbs 1E12 and 4B2 from ascites fluid were collected and purified. As shown in Figure 1D, the purified mAbs contained both heavy and light chains, with the molecular weight of the heavy chain being approximately double that of the light chain.

### 3.3. Immunogenicity and Specificity Assays of Monoclonal Antibodies

The specificity and immunogenicity of the mAbs 1E12 and 4B2 were assessed using Western blotting and IFA. Initially, the A29L protein was overexpressed in HEK-293T cells, and the samples were subjected to Western blot and IFA analyses. Both mAbs 1E12 and 4B2 specifically recognized the A29L protein, as depicted in Figure 2A,B. Furthermore, the results of the IP assays conducted on the HEK-293T cell lysates using the mAbs 1E12 and 4B2 demonstrate their ability to effectively enrich the A29L protein (Figure 2C).

We compared the amino acid sequence similarity between the MPXV A29L and VACV WR A27L proteins and found that they share 95% similarity (Figure 3A). Since studying MPXV requires biosafety level 3 (BSL-3) or higher laboratories, we used the VACV WR A27L protein, which has a high similarity to MPXV A29L, for testing. Subsequently, VACV WR was purified via sucrose gradient density centrifugation, yielding whole viral proteins. These proteins served as samples for the Western blot analysis using the acquired mAbs 1E12 and 4B2. As illustrated in Figure 3A, mAbs 1E12 and 4B2 demonstrated reactivity towards the A27L protein of VACV WR. To further assess the reactivity of mAbs 1E12 and 4B2 with VACV WR, IFA was performed using the prepared mAbs. Notably, a green fluorescence signal was observed in the infected cells treated with both mAbs, as well as anti-dsRNA (Orthopoxviruses produce large amounts of dsRNA during infection) [15], whereas the negative control showed no green fluorescence (Figure 3B). This finding demonstrates the strict specificity of hybridomas 1E12 and 4B2 towards VACV WR.

Following the infection of BHK-21 cells with VACV WR, the cells were collected and lysed. Subsequently, the antibodies were incubated with the cell lysates for 2 h to form antibody–protein A+G bead complexes. The proteins captured by these complexes were then eluted, separated via SDS-PAGE, and analyzed by Western blotting. The results reveal the presence of specific target bands in the lanes corresponding to VACV WR-infected cells, while no such bands were detected in lanes with uninfected cells. This observation signifies the ability of the monoclonal antibody to effectively enrich target proteins, as illustrated in Figure 3C.

### 3.4. A27L Protein Expression in VACV WR-Infected Cells at Different Time Points

VACV WR was used to infect BHK-21 cells, followed by IFA and Western blot detection at various time points postinfection, utilizing the mAbs 1E12 and 4B2 as primary antibodies. The IFA results reveal the presence of the VACV WR A27L protein as early as 6 h postinfection, as depicted in Figure 4. Notably, with prolonged VACV-WR infection, there was an increase in A27L protein expression accompanied by more severe cellular rounding. Overall, the findings from the IFAs highlight the early expression of the A27L protein during VACV WR infection, which persisted until cell death.

## 4. Discussion

Monkeypox virus was first discovered in laboratory monkeys in 1958. The first human case of monkeypox was identified in a 9-month-old child in the Republic of the Congo [16,17]. Since the initial case of monkeypox (MPX) in the current outbreak was documented on 7 May 2022, the disease has rapidly spread across numerous countries, including the UK, Spain, Portugal, Sweden, Belgium, Singapore, and South Korea. This spread represents a significant departure from its previous confinement, primarily to the rainforests of Central and West Africa [18]. Historically perceived as an easily overlooked ailment due to its sporadic occurrence and minimal attention in nonendemic regions, monkeypox has increasingly drawn concern as the number of confirmed cases escalates globally.

Orthopoxviruses have large and complex genomes, ranging from approximately 170 to 230 kb, and encode approximately 200 proteins [19]. The MPXV-A29L protein is an important structural protein of MPXV and a major target of neutralizing antibodies. Comparing the amino acid sequences of MPXV A29L and VACV A27L revealed significant similarities [20]. However, studies on MPXV must be conducted in a biosafety level 3 (BSL-3) laboratory. Consequently, in this study, we opted to detect the A27L protein in VACV WR-infected BHK-21 cells using hybridoma culture supernatants, facilitating the screening of mAbs.

Since the clinical symptoms caused by MPXV are difficult to distinguish from those caused by other poxviruses, diagnosing monkeypox based on clinical manifestations alone is challenging. Therefore, the rapid and accurate detection of MPXV is particularly important for epidemic prevention and control. Currently, laboratory detection methods for MPXV primarily include nucleic acid detection, immunological detection, and virus isolation and culture. Nucleic acid detection includes PCR, real-time fluorescence quantitative PCR (qPCR), loop-mediated isothermal amplification (LAMP), and whole-genome sequencing. Among these methods, the advantage of PCR and qPCR is that they can rapidly detect early infections in patients, but these methods can only detect nucleic acids from virus-infected individuals and cannot meet the needs of epidemiological investigations [21]. Although LAMP is easy to perform, it is prone to aerosol formation and cannot be used for large-scale testing [22]. Whole-genome sequencing can accurately determine the genus, species, and genetic evolution of viral strains and can identify genetic mutations, but it also cannot be applied on a large scale [23]. Virus isolation and culture require biosafety level 3 (BSL-3) laboratories and have high requirements for operators. mAbs against viral proteins are important biological materials for basic and applied research on viruses. The mAbs prepared in this study can be used for competitive ELISA, double-antibody sandwich ELISA, and other immunological detection methods targeting the A29L antigenic site. Compared with other antigen detection methods, these methods have the advantages of high sensitivity and strong specificity and do not require the antigen to be purified in advance.

In this study, the A29L protein of MPXV, which is expressed and purified from prokaryotic cells, was used as an antigen to immunize BALB/c mice to obtain mAbs. Two hybridoma cells lines capable of stably secreting A29L antibodies were screened using indirect ELISA and IFA with the cell fusion technique. The mAbs will provide new biomaterials for future research, the development of new diagnostic methods for MPXV, and the analysis of the role of A29L in MPXV.

## Figures and Tables

**Figure 1 viruses-16-01184-f001:**
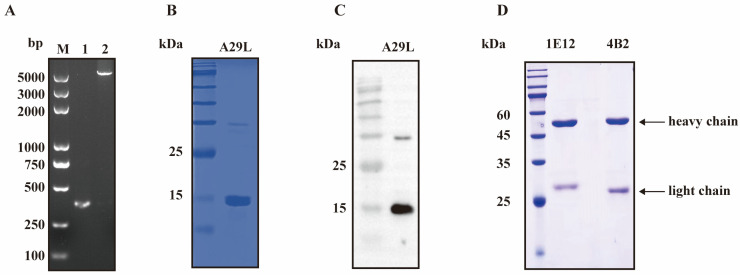
The expression and purification of the recombinant A29L protein and purification of the mAbs. (**A**) Lane 1: amplified products of the MPXV A29L gene. Lane 2: linearized vectors obtained by PCR amplification. (**B**) The purification of the recombinant A29L protein identified by SDS-PAGE. (**C**) A Western blot analysis of the recombinant A29L protein using an anti-His mAb. (**D**) The identification of the purified mAbs 1E12 and 4B2.

**Figure 2 viruses-16-01184-f002:**
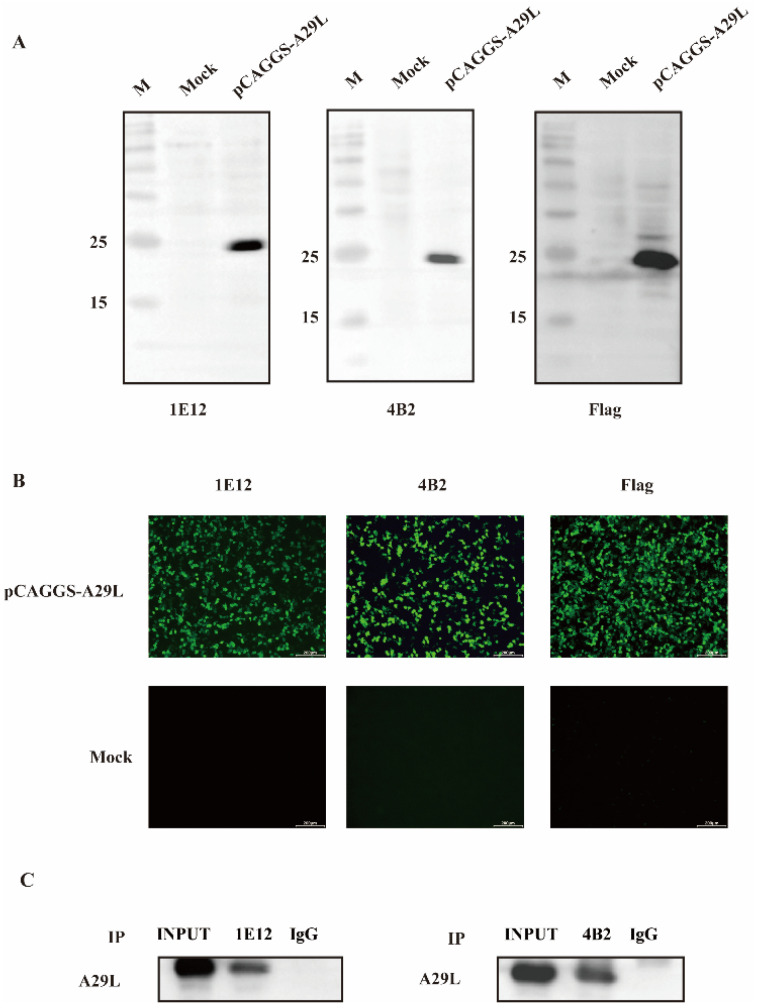
The specificity and immunogenicity of the mAbs 1E12 and 4B2 against the MPXV A29L protein strain were determined by Western blotting and IFA. (**A**) The Western blot detection of the A29L protein’s overexpression in HEK-293T cells using mAb 1E12, mAb 4B2, and anti-Flag mAb. (**B**) The IFA results of mAb 1E12 and mAb 4B2 in HEK-293T cells transfected with pCAGGS-A29L. (**C**) The lysates of HEK-293T cells were immunoprecipitated with either mAb 1E12 (left) or mAb 4B2 (right). The precipitates were analyzed by Western blotting using antibodies against Flag.

**Figure 3 viruses-16-01184-f003:**
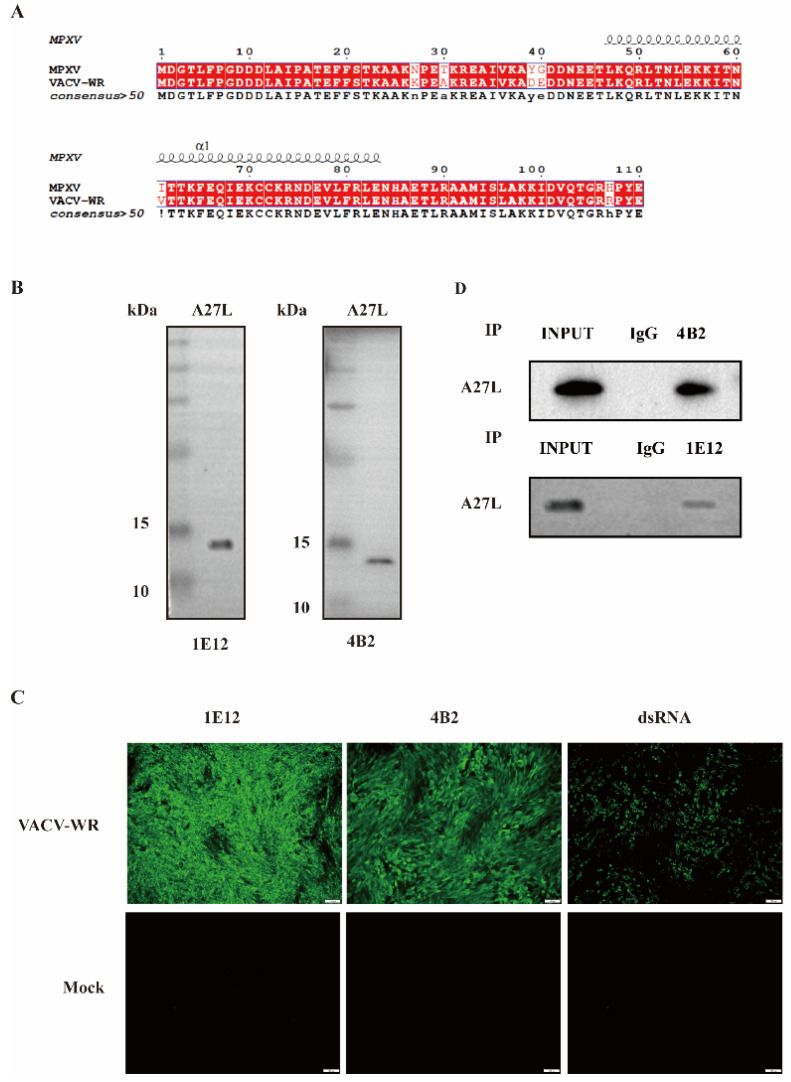
The cross-reactivity of mAbs 1E12 and 4B2 against the A27L protein of VACV. (**A**) A comparison of the amino acid sequence similarity between MPXV A29L and VACV WR A27L. (**B**) The Western blot detection of mAbs. The reactivity of mAb 1E12 and mAb 4B2 with the VACV WR A27L protein in infected cells was analyzed by Western blotting. (**C**) The IFA detection of mAb 1E12 and mAb 4B2 in VACV WR-infected cells. The green fluorescence represents the reaction of mAb 1E12 and mAb 4B2 with A27L in different strains of VACV. (**D**) The detection of the IP capacities of mAb 1E12 and mAb 4B2. The VACV WR-infected cells were lysed and immunoprecipitated with either mAb 1E12 or mAb 4B2. The precipitates were analyzed by Western blotting using antibodies against A27L.

**Figure 4 viruses-16-01184-f004:**
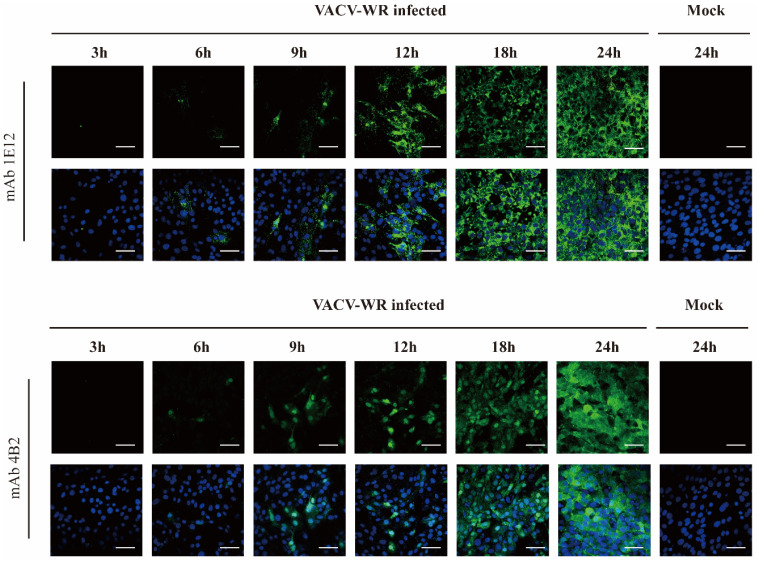
The detection of A27L protein expression at different time points after VACV infection. The expression of the A27L protein in VACV WR-infected cells was monitored at various time points postinfection using IFA. Scale bar, 50 μm.

## Data Availability

All data generated and analyzed during this study are included in this published manuscript.

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
