# Peer review of "The Generation and Characterization of Monoclonal Antibodies against the MPXV A29L Protein"

_viruses, 2024, doi:10.3390/v16081184_

Round 1
Reviewer 1 Report
Comments and Suggestions for Authors
The manuscript presents a scientifically rigorous study with a well-structured experimental design and coherent results. The research has the potential to make significant contributions to MPXV diagnostics. However, incorporating the suggested modifications and additional controls will further enhance the study, making it more robust and comprehensive.
Requests for Improvement/Corrections:
1. The construction of pET-A29L and pCAGGS-A29L should be detailed in the Methods section.
2. Figure 4: The DAPI chart for the Mock group appears to be misaligned.
3. Ethical Considerations: There is no mention of animal ethics in the current text. Please add a section addressing the ethical considerations regarding animal use.
4. Since monoclonal antibodies were used to demonstrate cross-reactivity with VACV A27L, the authors could have conducted a viral neutralization assay with the antibodies to assess their potential neutralizing/cross-neutralizing activity.
Author Response
Reviewer #2 (Comments and Suggestions for Authors):
The manuscript presents a scientifically rigorous study with a well-structured experimental design and coherent results. The research has the potential to make significant contributions to MPXV diagnostics. However, incorporating the suggested modifications and additional controls will further enhance the study, making it more robust and comprehensive.
Requests for Improvement/Corrections:
- The construction of pET-A29L and pCAGGS-A29L should be detailed in the Methods section.
We have added relevant content in the Materials and methods section. (Line 141 and Line 219)
- Figure 4: The DAPI chart for the Mock group appears to be misaligned.
We have realigned the DAPI chart for the Mock group to ensure consistency with the other charts in the figure.
- Ethical Considerations: There is no mention of animal ethics in the current text. Please add a section addressing the ethical considerations regarding animal use.
Thank you for pointing this out. We added a section on animal ethics in Line 155.
- Since monoclonal antibodies were used to demonstrate cross-reactivity with VACV A27L, the authors could have conducted a viral neutralization assay with the antibodies to assess their potential neutralizing/cross-neutralizing activity.
We have conducted further experiments to assess the neutralizing effects of the antibodies. We incubated 200 TCID50 units of virus with varying concentrations of antibodies at 37°C for 30 minutes. Subsequently, we inoculated the cells and established a virus control group. After a 24-hour incubation period, the cells were fixed using paraformaldehyde and subjected to immunofluorescence assays (IFA). Our findings indicate that neither of the two antibodies exhibited a neutralizing effect against the virus.

Reviewer 2 Report
Comments and Suggestions for Authors
The research design focuses on the generation and characterization of monoclonal antibodies (mAbs) targeting the MPXV A29L protein utilizing prokaryotic expression systems and hybridoma technology. This study aims to develop specific mAbs for detecting MPXV, thereby addressing a critical need for diagnostic and therapeutic purposes. With well-defined research objectives and a coherent progression from protein expression to antibody characterization, the study is effectively aligned with its goal of bridging the gap in specific diagnostic tools for MPXV.
Requests for Improvement/Corrections:
Line 160: "To obtain monoclonal cells." – Please clarify if this is a typographical error. Additionally, provide details on the purification process of the monoclonal antibodies (mAbs).
Line 183: Please include a description of the instruments utilized to observe the Immunofluorescence Assay (IFA) results.
Figure 4: Please specify the length of the scale, as it is currently not indicated in the diagrams. Providing the scale length is essential for accurate interpretation and comparison of the results.
The monkeypox virus A29L protein is relatively conserved within the genus Orthopoxvirus, and monoclonal antibodies targeting this protein exhibit extensive cross-reactivity. The two monoclonal antibodies prepared in this study, 1E12 and 4B2, using immunization with the A29L protein, also cross-react with the VACV-A27L protein. This raises the question of whether they also exhibit cross-reactivity with the A27L protein of other Orthopoxvirus members. This hypothesis can be tested by expressing proteins from other members of the genus Orthopoxvirus.
Minor editing of English language required.
Author Response
The research design focuses on the generation and characterization of monoclonal antibodies (mAbs) targeting the MPXV A29L protein utilizing prokaryotic expression systems and hybridoma technology. This study aims to develop specific mAbs for detecting MPXV, thereby addressing a critical need for diagnostic and therapeutic purposes. With well-defined research objectives and a coherent progression from protein expression to antibody characterization, the study is effectively aligned with its goal of bridging the gap in specific diagnostic tools for MPXV.
(1) Line 160: "To obtain monoclonal cells." – Please clarify if this is a typographical error. Additionally, provide details on the purification process of the monoclonal antibodies (mAbs).
Response: We have made the corresponding changes in the original text and marked them in red font, and added the details of the purification process of the mAbs in Line 165.
(2) Line 183: Please include a description of the instruments utilized to observe the Immunofluorescence Assay (IFA) results.
Response: We have made the corresponding changes in the original text and marked them in red font in Line 189.
(3) Figure 4: Please specify the length of the scale, as it is currently not indicated in the diagrams. Providing the scale length is essential for accurate interpretation and comparison of the results.
Response: Thank you for pointing this out. We added a note about the ruler length in the legend in figure 4.
(4) The monkeypox virus A29L protein is relatively conserved within the genus Orthopoxvirus, and monoclonal antibodies targeting this protein exhibit extensive cross-reactivity. The two monoclonal antibodies prepared in this study, 1E12 and 4B2, using immunization with the A29L protein, also cross-react with the VACV-A27L protein. This raises the question of whether they also exhibit cross-reactivity with the A27L protein of other Orthopoxvirus members. This hypothesis can be tested by expressing proteins from other members of the genus Orthopoxvirus.
Response: Following your guidance, we conducted a thorough comparison of the amino acid sequences. Our analysis revealed that the sequences of A27L from Cowpox, Variola, A29L from Monkeypox virus (MPXV), and A27L from Vaccinia virus (VACV) share an impressive identity of 94.55%. This high degree of similarity led us to hypothesize that these proteins could specifically interact with the A27L protein found in Cowpox and Variola. To test this hypothesis, we constructed overexpression plasmids, specifically pCAGGS-A27L (Cowpox) and pCAGGS-A27L (Variola). Utilizing immunofluorescence assays (IFA), we confirmed that antibodies 1E12 and 4B2 exhibit a specific reaction with these proteins. The following are the A27L amino acid sequence comparison results and IFA results.
